# A Highly Sensitive Molecular Technique for RNA Virus Detection

**DOI:** 10.3390/cells13100804

**Published:** 2024-05-09

**Authors:** Tomasz Rozmyslowicz, Haruki Arévalo-Romero, Dareus O. Conover, Ezequiel M. Fuentes-Pananá, Moisés León-Juárez, Glen N. Gaulton

**Affiliations:** 1Department of Pathology and Laboratory Medicine, Perelman School of Medicine, University of Pennsylvania, Philadelphia, PA 19104, USA; dconover@sas.upenn.edu (D.O.C.); gaulton@pennmedicine.upenn.edu (G.N.G.); 2Laboratorio de Inmunología y Microbiología Molecular, División Académica Multidisciplinaria de Jalpa de Méndez, Departamento de Genómica, Universidad Juárez Autónoma de Tabasco, Jalpa de Méndez 86205, Mexico; haruki.arevalo@ujat.mx; 3Unidad de Investigación en Virología y Cáncer, Hospital Infantil de México Federico Gómez, Ciudad de México 06720, Mexico; ezequiel.fuentes@alumni.bcm.edu; 4Laboratorio de Virología Perinatal, Departamento de Inmunobioquímica, Instituto Nacional de Perinatología Isidro Espinosa de los Reyes, Ciudad de México 06720, Mexico; moisesleoninper@gmail.com

**Keywords:** Zika, Chikungunya, LAMP, detection, diagnostics

## Abstract

Zika (ZIKV) and Chikungunya (CHIKV) viruses are mosquito-transmitted infections, or vector-borne pathogens, that emerged a few years ago. Reliable diagnostic tools for ZIKV and CHIKV—inexpensive, multiplexed, rapid, highly sensitive, and specific point-of-care (POC) systems—are vital for appropriate risk management and therapy. We recently studied a detection system with great success in Mexico (Villahermosa, state of Tabasco), working with human sera from patients infected with those viruses. The research conducted in Mexico validated the efficacy of a novel two-step rapid isothermal amplification technique (RAMP). This approach, which encompasses recombinase polymerase amplification (RPA) followed by loop-mediated isothermal amplification (LAMP), had been previously established in the lab using lab-derived Zika (ZIKV) and Chikungunya (CHIKV) viruses. Crucially, our findings confirmed that this technique is also effective when applied to human sera samples collected from locally infected individuals in Mexico.

## 1. Introduction

In recent years, the need for rapid decision-making in the triage of febrile illness due to infectious diseases has grown. This has led to the development and testing of a low-cost, fast (<40 min), high-sensitivity, compact, and field-deployable multiplexed device for simultaneous molecular detection of the blood-borne pathogens such as the Zika virus (ZIKV), Chikungunya virus (CHIKV), dengue virus (DENV), and many others is growing [1,2,3]. The potential for infection and/or co-infection with these agents occurs in remote regions of the world with limited access to medical diagnostic services. Rapid diagnosis of the cause of illness by infectious agents that have overlapping early clinical symptoms is essential to determine whether treatment or evacuation is required. Moreover, co-infections can alter symptoms and affect therapy. For example, regions endemic with ZIKV are also susceptible to CHIKV, DENV, and many other viruses—all resulting in similar initial symptoms but require diverse disease management strategies [4].

These acute viral (RNA virus) infections pose a considerable threat to the general public. This is particularly evident in the Middle East and Africa and lately in South and Central America, where multiple debilitating vector-borne diseases are endemic and can be readily acquired through the bite of infected arthropods such as mosquitos and ticks. Infections by these agents (e.g., DENV, ZIKV, CHIKV, and others) have been classified by the WHO as major health problems in these and other locations [3]. In each instance, effective vaccines and drug prophylaxis are not yet available.

Reliable diagnostic tools for the agents that cause these diseases are vital for appropriate disease management and therapy to prevent early misdiagnosis that may worsen clinical consequences and/or spread and for tracking transmission and resistance patterns. Portable diagnostic tools for these agents are not currently available for use in the point-of-care (POC) setting [5]. While our immediate objective is to develop a POC molecular diagnostic platform for the detection of these pathogens, our long-term objective is to leverage the multiplexed capacity of this device for the simultaneous detection of many other distinct pathogens.

Zika virus (ZIKV), belonging to the *Flaviviridae* family, was identified in Uganda in 1947 but only studied intensively after the virus spread in the Americas in 2015. Three structural and seven non-structural proteins are encoded in the ZIKV (+) ssRNA genome which has a size of ~11 kb. The virion contains the structural proteins prM/M and E in the nucleocapsid, arranged with icosahedral symmetry [6].

The life cycle of ZIKV follows a mechanism that is broadly similar to that of other viruses in its family, including dengue and West Nile. This process begins when viral particles bind to attachment factors on the surface of susceptible cells, such as monocytes, skin dendritic cells, neurons, Hoffbauer cells, trophoblasts, and testicular cells. The virions specifically interact with different receptors on these cells, including C-type lectin CD209 antigen (also known as DC-SIGN), T cell immunoglobulin mucin domain protein 1 (TIM-1), tyrosine-protein kinase receptor 3 (TYRO3), and AXL. For the virus to release its RNA into the cytoplasm, the particles must be internalized through a clathrin-dependent entry pathway. This pathway facilitates exposure of the viral envelope glycoproteins to an acidic environment, promoting conformational changes essential for the fusion of endosomal and viral membranes, which in turn releases the viral genome [7].

Inside the rough endoplasmic reticulum, positive-sense RNA is recognized by ribosomes that initiate the translation of a single polyprotein. This polyprotein is subsequently cleaved into ten proteins: three structural proteins (capsid, precursor membrane (prM), and envelope (E)) and seven non-structural proteins (NS1, NS2A, NS2B, NS3, NS4A, NS4B, and NS5) that are required for the replication of viral RNA. Subsequently, the viral replicase complex generates new RNA (+), which is then encapsulated into new viral particles within the lumen of the endoplasmic reticulum (ER). these newly formed viral particles mature and exit from the cell through the conventional secretory pathway from the infected cells [8].

The syndrome caused by ZIKV in humans was initially classified as a mild self-limiting febrile illness, resolved in infected individuals within only a few days. However, during the most recent epidemic, it has emerged as a global health concern due to its association with severe neurological defects in newborns [9,10,11], including microcephaly and Guillain–Barré syndrome (GBS), multi-organ failure, thrombocytopenia, and thrombocytopenic purpura [12,13]. Hence, the epidemic stimulated the development of vaccines and therapeutics, as well as models to understand the pathogenesis. Although it has been shown that neutralizing antibodies provide protection against the virus, no vaccines or treatments are currently approved for the treatment of ZIKV infection [14].

ZIKV transmission can occur through *Aedes* mosquito vectors, sexual contact, maternal–fetal transmission, and blood transfusions. Moreover, ZIKV infiltrates several biological fluids such as breastmilk, urine, tears, and saliva [15]. The incubation period is typically 3–12 days, but men can harbor infectious viruses in their testes for weeks. An ultrasound study of 88 ZIKV-positive pregnant women revealed serious developmental abnormalities in 29% of these instances [9]. The CDC strongly recommends screening pregnant women residing in or traveling to areas with ongoing ZIKV transmission.

Chikungunya virus (CHIKV), a (+) ssRNA arbovirus from the *Togaviridae* family presents clinical symptoms similar to dengue and certain aspects of ZIKV and is also transmitted by *Aedes* mosquito vectors, primarily *Aedes aegypti* but also *Aedes albopictus* [11,16].

CHIKV is a spherical enveloped virus of approximately 70 nm in diameter with a genome size of approximately 12 Kb. It encodes proteins via two open reading frames (ORFs): 5′ORF and 3′ORF, which are translated from genomic and subgenomic RNA, respectively. 5′ORF produces the non-structural polyprotein P1234, which is further cleaved into proteins nsP1 to nsP4. Meanwhile, 3′ORF is responsible for encoding structural proteins: capsid (C), envelope 3 (E3), envelope 2 (E2), 6K, and envelope 1 (E1) [17,18]. Upon interaction with target cells, the protein E2 binds to the Mxra8 receptor, triggering clathrin-mediated endocytosis. Subsequent endosomal acidification induces a rearrangement in the E1–E2 heterodimers, exposing the fusion loop in the E1 protein that allows for the fusion of the endosomal and viral membranes. Additionally, other molecules such as glycosaminoglycans (GAGs), TIM, DC-SIGN, AXL, and CD147 have been identified as alternative receptors for CHIKV [19,20,21].

After membrane fusion, the viral nucleocapsid is released into the cytoplasm, and the genomic RNA is translated into polyprotein 1234. This is processed to release the viral polymerase nsP4, which synthesizes a negative-strand RNA serving as a template to create new positive-strand RNA. Subsequent steps involve the synthesis of structural proteins beginning with the capsid protein C, which plays a critical role in packaging the viral RNA into nucleocapsids through oligomerization [22]. In the endoplasmic reticulum (ER) and Golgi apparatus, the pE2-6k-E1 polyprotein undergoes processing by cellular proteases. This leads to post-translational modifications, producing the structural proteins E1, E2, E3, and 6K. Glycosylation of E1 and E2 enables them to form heterodimers that will be used in the viral envelope and that are essential for generating infectious particles. Ultimately, the nucleocapsid core associates with membrane regions enriched with E1-E2 dimers, culminating in the release of mature virions from the infected cell [23].

In the tropics of Asia and Africa, the virus has been known to cause the disease for more than 60 years and only recently became a problem in developed countries. Outbreaks of Chikungunya disease occurred from Kenya to India in 2004, then in Italy in 2007, and finally in the Americas in 2014 [24]. There is a rural cycle involving mosquitos, nonhuman primates, and other mammals as well as an urban cycle involving mosquitos and humans. In rural areas, the disease is endemic, but in urban areas, outbreaks are sporadic but capable of infecting a large portion of susceptible populations [25].

Both sexes and all ranges of age are susceptible to the infection, which is rarely life-threatening and usually self-limiting. In almost half of adults, skeletal symptoms are persistent. The most sensitive population is infants due to complications in the central nervous system that have long-term consequences. Past outbreaks suggest that CHIKV may also establish persistent infections and long-term arthralgia [26].

The vertical transmission of CHIKV has not been reported, although transmission via infected blood products and organ donation is observed [27]. CHIKV disease typically features the sudden onset of fever, joint pain, and rash 3–7 days post exposure, typically lasting 2 to 10 days [28]. The fever rises rapidly and corresponds to the period of viremia (≥10^6^ units (pfu)/mL) when monocytes are infected and produce large quantities of type I IFN. Joint pain appears suddenly and can be incapacitating. Pain may persist many months after the original illness [29]. Since the 2013 outbreak in the Americas, CHIKV has spread across the Western Hemisphere and globally, with infections detected in 45 countries, as well as multiple US states [27].

Currently, there are no approved cross-species/type vaccines for any of these pathogens. Our decision to focus on point-of-care (POC) diagnosis for ZIKV and CHIKV is influenced by the severity of disease outcomes, the potential global health impact, the lack of effective vaccines, and the current limitation of diagnostic assays to professionally staffed laboratories. POC tests for ZIKV or CHIKV are not currently available. While the FDA has issued emergency use authorization for several laboratory-based ZIKV diagnostic tests, including RT-PCR assay and Zika MAC-ELISA16, most vendors provide limited details, making it challenging to compare test efficacies [30,31].

Our approach is simple, inexpensive, disposable, and incorporates an array of (RT)-LAMP chambers. This system differs from conventional POC molecular tests by incorporating a nucleic acid (NA) capture membrane at the inlet of the amplification chambers, which immobilizes NA, serving as templates for amplification [32]. In addition to isolating NAs, the membrane decouples the sample volume from the reaction volume, removing limitations on sample volume, and greatly improving sensitivity over typical rapid molecular tests. The device also features an efficient, high-capacity plasma separator that does not require electricity or centrifugation. Thermal control for the amplification reactions is provided either with a battery-powered thin film heater or by an electricity-free, water-activated exothermic chemical reaction [33,34,35]. The amplicons can be detected with intercalating fluorescent dye, bioluminescence, visible indicators (e.g., leuco triphenylmethane LCV), or pH. Notably, the fluorescence can be excited, monitored, analyzed, and transmitted using a cell phone [34]. This system is sensitive, capable of detecting <10 RNA copies in blood and saliva and 2 × 10^−17^ g/μL *S. mansoni* DNA in serum, and has been used for genotyping malaria-transmitting mosquitoes [32,36,37].

Building upon these advancements, we propose a novel, two-stage isothermal-isothermal amplification protocol, integrated into a next-generation chip. The first stage amplifies all targets in the sample. These targets then diffuse into secondary reactors for specific amplification. This innovative design facilitates the detection of multiple targets and co-infections with high sensitivity and specificity. The focus of this study lies in primer design, assay optimization, and the testing of two-step protocols on the benchtop with ZIKV and CHIKV, collected from infected local individuals in rural Mexico.

Generally, immune assays are much less sensitive and specific than nucleic acid-based tests. Nucleic acid tests (NAT) amplify a conserved sequence of the target’s nucleic acid, offering high specificity (no cross-reactivity), and sensitivity. However, these tests are often reliant on RT-PCR methodology, which demands extensive sample prep, expensive equipment, and highly trained personnel. Furthermore, NATs also necessitate knowledge of the target sequences.

During the last ZIKV outbreak, viruses isolated from patients in Surinam and Brazil exhibited high similarity to the strain that circulated in French Polynesia in 2013, with over 99.7% and 99.9% identity in nucleotide and amino acid sequences, respectively [38]. This high degree of similarity suggests that well-designed primers could provide a highly effective test.

RT-PCR is effective in detecting ZIKV in serum and saliva up to 14 days post-infection and for even longer periods in urine, whole blood, and semen. CHIKV, on the other hand, is detectable in serum and saliva but not urine [39,40,41,42]. Notably, a recent study discovered detectable ZIKV in pregnant women throughout their pregnancy [43].

Several companies are developing multiplexed RT-PCR kits to co-detect a variety of mosquito-borne pathogens. However, all available kits require a professional laboratory setting. In an interesting development, researchers have recently combined NASBA with a novel toehold switch to detect ZIKV in a low-cost, paper-based format [44,45]. Although refined, this method requires sample preparation and offers a relatively low limit of detection 31 of ~106 copies/mL in more than 3 h. This sensitivity is two orders of magnitude less than that offered by the CDC-recommended Trioplex real-time RT-PCR assay.

In contrast, our proposed method incorporates sample preparation and provides a superior sensitivity of one PFU per sample in less than 40 min. Given the expected rise in ZIKV sexual transmission from asymptomatic individuals returning from endemic regions, there is a significant need for a highly sensitive ZIKV test. Immune assays, due to their inferior sensitivity and specificity compared to nucleic acid-based tests, fall short of this requirement.

## 2. Materials and Methods

### 2.1. Clinical Samples

This study incorporated a total of 38 clinical samples, specifically sera. Out of these, 19 samples were sourced from ZIKV-infected patients, while the remaining 19 were obtained from healthy donors. Initially, samples were collected in hospitals in the state of Tabasco Mexico from symptomatic patients who had been confirmed as ZIKV positive via RT-PCR in the Laboratorio Estatal de Salud Publica del Estado de Tabasco during the years 2016–2018. The positive samples were procured from this laboratory and these serum samples were stored at −20 °C until direct dilution for RPA-LAMP reactions. The acquisition and use of human samples were reviewed and approved by the University of Pennsylvania Institutional Review Board (IRB)—protocol 809496

Due to multiple technical challenges in obtaining CHIKV-positive sera samples from the same source, and other government institutions (either local or federal), we opted to use a laboratory isolate of Chikungunya virus from an infected patient, acquired from our collaborator in Mexico City (Instituto Nacional de Perinatología INPER). This collaborator also supplied us with a ZIKV-positive control of cDNA, which was essential for our subsequent studies, especially gel electrophoresis.

### 2.2. Recombinase Polymerase Amplification-Loop-Mediated Isothermal Amplification Primer Design

Previously, our group used GenBank data for genomic sequences of ZIKV/CHIKV strains that were isolated in different places in Latin America (Brazil, Mexico, Suriname, and Columbia) to design primers. Using the DNAMAN software, these sequences were aligned and compared with sequences of the dengue virus to obtain specific primers. The region chosen for the primer design was the envelope protein because it has high homology with the different isolated ZIKV strains but differs from the dengue strains. Hence, we identified highly conserved and unique sequences within the RNA of ZIKV and CHIKV.

These sequences do not occur in other organisms, mitigating the risk of cross-reactivity. We designed the primer sets using the PrimerExplorer V4 software available from Eiken Chemical Co., Ltd., Tokyo, Japan, and these were synthesized by IDT, in Coralville, IA, USA.

### 2.3. Recombinase Polymerase Amplification-Loop-Mediated Isothermal Amplification Reaction

The LAMP method is based on auto-cycling strand displacement DNA synthesis using a set of inner and outer primers that allow for the recognition of six different sequences, ensuring high specificity for target amplification. The inner primers are called the FIB (forward inner primer), BIP (backward inner primer), Loop F, and Loop B, respectively. The outer primers consist of the B3 and F3 sequences. The LAMP reaction is initiated with a cycle for denaturing the DNA sample and then carried out at 60–65 °C for 1 h. The final products of LAMP are a mixture of stem-loop DNAs of various lengths and cauliflower-like structures formed by self-complementary sequences that alternate between inverted repeats on the same strand of the target sequence [46].

On the other hand, recombinase polymerase amptlification relies on recombinase and displacing DNA polymerase proteins. The process begins with the formation of a recombinase-primer complex in the presence of ATP. This complex then locates the homologous sequence in the target, promoting strand invasion. Finally, amplification is carried out using a strand-displacing DNA polymerase at temperature ranges from 37–42 °C, completing within 20–40 min [47].

Our two-stage isothermal RPA-LAMP assay relies on the two critical phases mentioned before: the initial RPA reaction uses the outer forward (F3) and backward (B3) primers for each target. Then, these first-stage amplicons serve as templates for the highly specific second-stage LAMP reaction.

In the first stage, the RPA master mix TwixAmp (TwistDx, Cambridge, UK) reaction consisted of 29.5 µL of rehydration buffer, 2.4 µL (10 µM) of F3 and B3 primers, 12.2 µL of target serum, 1 µL AMV reverse transcriptase (2 U PROMEGA, Madison, WI, USA), and 2.5 µL of magnesium acetate (280 mM). This first reaction was carried out for 20 min at 38 °C on a Thermal Cycler ProFlex PCR system (Applied Biosystems, Foster City, CA, USA).

For the second stage, 1 µL of the RPA reaction was used in an OptiGene Isothermal Master Mix ISO-DR001 (OptiGene, Horsham, UK). This mix included: 15 µL of OptiGene mix, 1.3 µL of a mix of primers (1.6 μM FIP and BIP; 0.8 μM LF and LB), and 7.7 µL of water. The amplification was carried out and monitored with a Peltier Thermal Cycler PTC-200 (Bio-Rad DNA Engine, Hercules, CA, USA) at 60 °C. Fluorescence emission intensity data were collected once every minute for 60 min. For these experiments, we used sera from infected patients (see above) along with sera from healthy individuals as a negative control, and laboratory-isolated CHIKV in 3 different concentrations.

### 2.4. Confirmation via Gel Electrophoresis

Following the LAMP reaction, some samples were divided in half and were electrophoresed in 1% agarose gel. 1× TAE buffer (10 mM Tris, 20 mM acetic acid, and 1 mM EDTA) was used at 75 V for 60 min. The gels were visualized under UV light using the Enduro ^TM^ GDS imaging system (Labnet International, Inc., Edison, NJ, USA). This study, however, was conducted exclusively for ZIKV samples due to challenges encountered in acquiring sera from patients infected with CHIKV.

### 2.5. Statistical Analysis

Data are presented as the mean ± standard deviation. For comparisons between the data groups, Student’s *t*-test was applied. A *p*-value equal to or lower than 0.05 was used to establish significance between groups.

## 3. Results

Figure 1 shows the protocol followed in this study. It can be seen that after obtaining the sample and separating the serum, the RPA and LAMP reactions can be carried out as quickly as one hour.

Our first objective was to assess the functionality of our method using sera samples from infected patients. In healthy donors’ serum (used as a negative control), fluorescence intensity increase was notably late during the incubation reactions (Figure 2, cycle 49). However, when using sera from ZIKV-infected patients, the intensity curves peaked at early incubation reaction cycles, indicating efficient virus detection (Figure 2, cycles 10–19). Samples 03 (cycle 10) and 08 (cycle 13) exhibited readily discernible fluorescence intensity, while samples 11, 15 (cycle 16), and 05 (cycle 19) were positive, albeit with reduced intensity.

To validate the detection of our samples, we measured the fluorescence intensity as a function of time throughout the amplification reactions. Figure 3 shows that samples 19, 17, and 06 exhibited an earlier fluorescence increase than sample 14, despite them all being positive. The negative control serum remained at the fluorescence baseline over the entire 60-min incubation time.

Finally, the relative expression levels of the ZIKV target were represented using the 2^−ΔΔCt^ equation, setting control group samples to 1.0. As shown in Figure 4, compared to the control, the detection of ZIKV samples increased over 1 × 10^4^-fold. These results confirmed the efficacy of our method in detecting the ZIKV virus in the human sera and suggest a correlation between high target concentrations and early detection cycles/times.

To confirm our RPA-LAMP findings, we conducted an additional 2% agarose gel analysis with the samples described above. We analyzed the patient samples previously identified as ZIKV-positive by our RPA-LAMP using 2% agarose gel for final confirmation. Figure 5 displays randomly selected patient samples (6 in total: Z-03, Z-08, Z-11, Z-14, Z-15, and Z-05) alongside our positive control (ZIKV cDNA as above) and negative control (Neg-S-02). All patient samples appeared to be positive on the 2% agarose gel, validating the utility of our technique of RPA-LAMP.

To ensure the applicability of this technique to CHIKV, we performed RT-PCR (RPA-LAMP) on Chikungunya virus samples obtained from our collaborators in Mexico, in three different concentrations. The positive results, as depicted in Figure 6, verify our method’s potential for the fast detection of RNA virus infection.

## 4. Discussion

During the recent outbreaks of ZIKV in the Americas, the most alarming characteristic of the infection was the induction of microcephaly, congenital malformations, fetal demise, and also the development of Guillain–Barré syndrome [9]. Similarly, during the 2013 CHIKV outbreak in the Americas, classic CHIKV symptoms, such as fever and severe joint pain of varying intensity and duration, were reported [7,8]. The re-emergence of these viral outbreaks resulted in the widespread distribution of these viruses throughout the Americas, with them becoming endemic diseases.

One report, combining metagenomic sequencing with phylogenetic, epidemiological, and environmental data, details the spread of ZIKV across Central America and Mexico. The findings showed multiple ZIKV introductions into Central America, with a single lineage originating in Brazil becoming dominant and spreading through Honduras due to its high suitability for the ZIKV mosquito vector. The evidence suggests a complex annual transmission trend in 2016 with two distinct waves. The first wave occurred during spring and summer in Mexico and Nicaragua, while the second took place during the winter in Honduras, Guatemala, and El Salvador [48].

In the case of CHIKV, the spread exhibited similarities to ZIKV. One report shows thirteen introductions for the virus into Mexico, with half of them leading to persistent transmission. However, CHIKV only evidenced a single transmission event from the Asian lineage, which was then followed by the virus’s spread across the Central Caribbean and South, Central, and North America [49].

In Mexico, ZIKV surveillance has been carried out since 2015 through the Mexican Social Security Institute (IMSS). Their 2019 report detailed the results of this epidemiological monitoring, revealing a total of 13,487 positive ZIKV cases, with the majority being women (63%). Every state in the country reported suspected ZIKV cases, with the bulk of them occurring in the southeastern region. Additionally, the monitoring of positive cases in pregnant women showed 1082 positive cases for ZIKV (32%) via RT-PCR [50].

Regarding CHIKV, monitoring in three regions of southwestern Mexico (Chiapas State) using RT-PCR and ELISA revealed that 80% of reported cases of febrile illness and polyarthralgia were due to CHIKV infection. The circulating strain was related to the Asian lineage, aligning with several reports [51].

These variations in viral spread across the Americas underline the need for real-time surveillance methods to enable timely intervention responses. This is a significant challenge, considering that adult infections often result in non-specific, flu-like symptoms, similar to those caused by another endemic infectious agent, the dengue virus. Such symptom similarity makes misdiagnosis among these three viruses a real possibility.

Molecular detection offers an advantage over clinical symptom-based diagnosis, and the gold standard for such diagnosis is RT-PCR. While this method is both sensitive and specific, it usually requires skilled personnel and specialized laboratory infrastructure. These requirements result in time-consuming processes and elevated costs. For example, 786 (23.1%) samples of the study of ZIKV could not be processed due to laboratory rejection criteria for biological samples, such as lipemic, contaminated, or hemolyzed samples. Other reasons included insufficient sample volume for RT-PCR or samples that were not constantly kept cold prior to analysis [50]. Therefore, the development of diagnostic options that are fast, reliable, and easy to execute becomes critical. In recent years, our working group has focused on the development of point-of-care (POC) molecular diagnostics to overcome resource constraints and improve healthcare [32,37,52].

POC diagnostics have several advantages for detecting infectious diseases: they are easy to use and analyze, cost-effective (the cost of the assay according to our estimation is less than or equal to USD 10.00 per sample), quick, and require little if any laboratory infrastructure without compromising on sensitivity or specificity. Moreover, they can be deployed in hard-to-reach areas, such as endemic regions. RT-LAMP has been one of the most valuable techniques for screening and testing for viruses and other infectious diseases, and our group has used it extensively. Previously, working with ZIKV, we applied a POC device that consisted of a microfluidic cassette with independent multifunctional amplification reactors. This approach eliminated the need to isolate RNA from the samples, allowing for their direct use in RT-LAMP assays. Sensitivity was tested with saliva samples spiked with varying concentrations of ZIKV and compared with benchtop RT-LAMP [52]. This study aims to improve upon this by employing a novel two-step rapid isothermal amplification (RAMP) technique or recombinant polymerase amplification (RPA), followed by loop-mediated isothermal amplification (LAMP), using human serum samples obtained from infected individuals.

In our examination of ZIKV clinical samples, all previously tested positive cases were confirmed as infected via the RPA-LAMP method. Interestingly, 83% of the ZIKV-positive samples were from female patients, which aligns with the findings in the IMSS study in 2019, reporting 63% of positive samples from female patients and only 16% from male patients [50]. All these samples were obtained from patients between 29 and 38 years of age. In the case of ZIKV, we further validated some of the positive samples via gel electrophoresis. Lastly, all 20 control samples were negative in specific RAMP reactions.

While other research groups have also developed new technologies for viral disease detection, many have only been tested on spiked samples. Some require the prior isolation of nucleic acids and have not been tested with human samples that could contain inhibitors for the RT-LAMP reaction. For example, by using automated magnetic particle (MP)-based nucleic acid extraction with rapid real-time RT-RPA or RT-PCR, one study showed the detection of ZIKV in 30 min. However, the urine samples were spiked with a virus produced in the laboratory [53]. Another study showed that by using a kit with Q-paper and RT-LAMP, the detection of ZIKV and CHIKV can be read in 30 min. Nonetheless, for ZIKV, they also used viruses propagated in the laboratory, and the methodology includes prior isolation of the viral RNAs [54]. Finally, other studies using simulated clinical samples with ZIKV showed the detection of the virus via RT-LAMP but with pretreatment of the samples [55].

In this study, we have demonstrated that our novel two-step isothermal amplification assay performs effectively on endogenously infected samples, not merely laboratory-derived ones. It holds potential for future usage in POC multiplatform devices for rapid virus detection. Importantly, as our previously developed POC device operates without electricity, it substantially reduces the time and cost of assays in endemic areas like Tabasco, Mexico.

## 5. Conclusions

In conclusion, we have developed a fast and robust assay for detecting ZIKV and CHIKV in patients’ infected sera. The simplicity of our RPA-LAMP assay suggests that it can be used in endemic areas without special training, offering a new diagnostic tool for monitoring outbreaks of these viruses. We are also working to expand our system to detect additional targets, such as dengue virus (DENV) and other RNA viruses.

## Figures and Tables

**Figure 1 cells-13-00804-f001:**
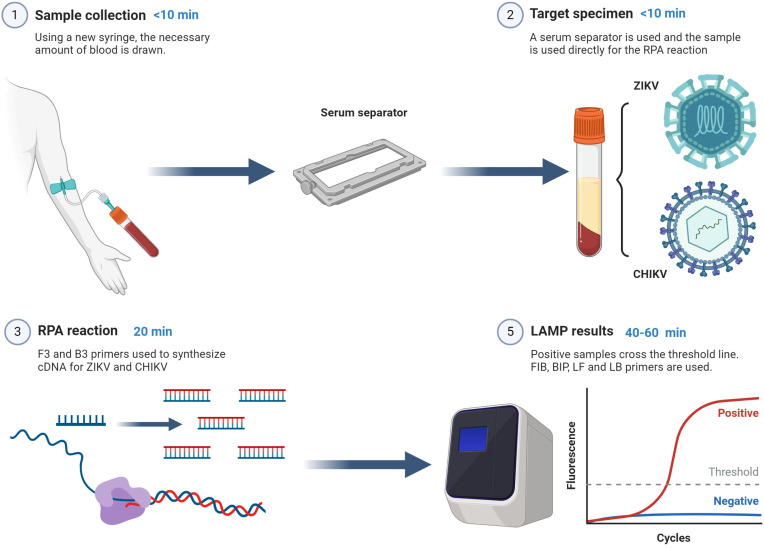
Graphical representation of the protocol used in this study. This figure was created with BioRender.com.

**Figure 2 cells-13-00804-f002:**
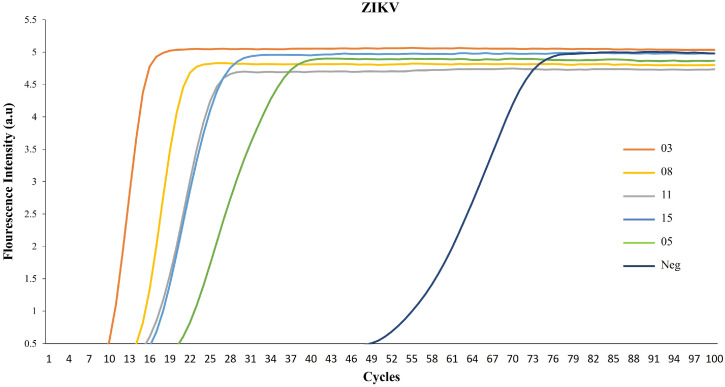
RPA-LAMP amplification curves from ZIKV samples. Real-time monitoring of LAMP amplification as a function of cycles. Five samples of ZIKV and one negative serum are shown.

**Figure 3 cells-13-00804-f003:**
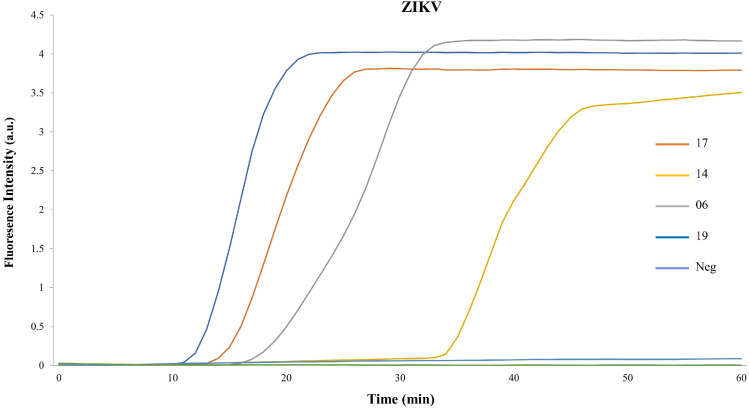
RPA-LAMP amplification curves from ZIKV as a function of time. The real-time monitoring of LAMP amplification as a function of time is shown. Four ZIKV samples and negative serum are shown.

**Figure 4 cells-13-00804-f004:**
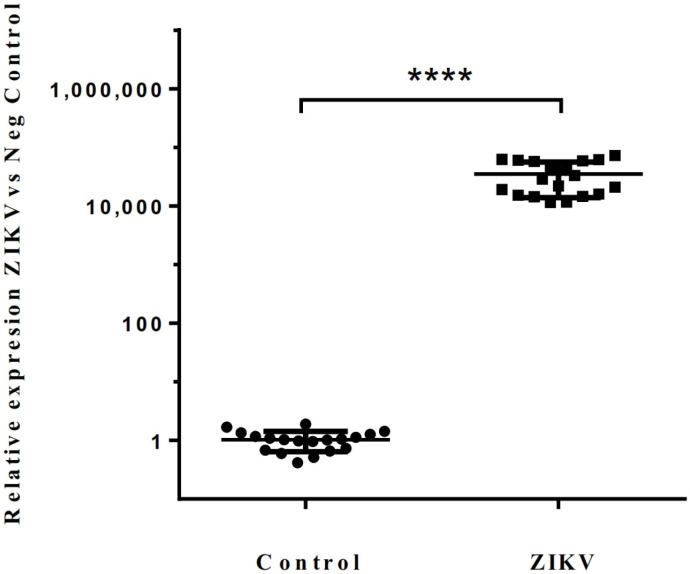
Relative detection of ZIKV samples. Data are presented as individual values with the median and range from independent experiments using all 19 samples from ZIKV. Negative serum samples were set at 1.0 to show the fold in the detection. The asterisk represents a *p*-value equal to 0.0001 relative to the negative serum samples.

**Figure 5 cells-13-00804-f005:**
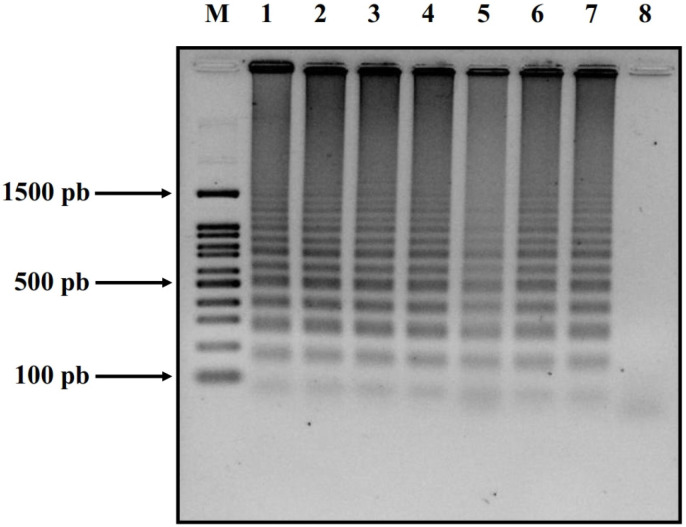
Confirmation of ZIKV in the clinical samples. ZIKV clinical samples subjected to RPA-LAMP reactions visualized using 2% agarose gel. The ZIKV selected patients’ samples: line 1 (ZIKV cDNA control), line 2 (Z-03), line 3 (Z-08), line 4 (Z-11), line 5 (Z-14), line 6 (Z-15), line 7 (Z-05), and line 8 (Neg-S-02). M—100 pn DNA ladder.

**Figure 6 cells-13-00804-f006:**
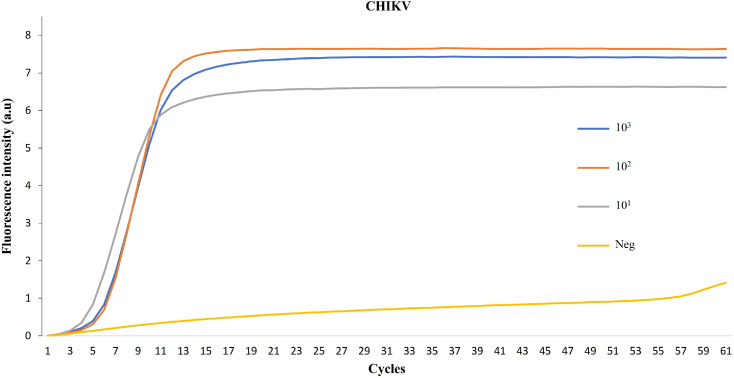
Sensitivity assessment of CHIKV. Using three serial dilutions of the CHIKV standard RPA-LAMP reactions were developed. The real-time monitoring is shown as a function of cycles.

## Data Availability

The raw data supporting the conclusions of this article will be made available by the authors on request.

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
