# Peer review of "A Highly Sensitive Molecular Technique for RNA Virus Detection"

_cells, 2024, doi:10.3390/cells13100804_

Round 1

Reviewer 1 Report

Comments and Suggestions for Authors

The search of simplified methods for diagnostic of infectious diseases is very important, especially for low-income countries. However, the proposed methodology should be describe more detail. A photo of the simple equipment used should be given. It seems that the clinical samples could be more quantity. In the section "Discussion" should be add another alternative methods of diagnostic of viral infection for compated with proposed.

Author Response

We really appreciate your time to review our manuscript. As suggested in this review, we added more details to describe the methodology used.

Please see the prototype (attached as PDF file) of the POC device designed at the University of Pennsylvania, School of Engineering and Applied Sciences (courtesy of Dr. Haim H. Bau).

We think, however, that including this photo now in our paper would be premature since some parts of this device are still under patent review.

We strongly agree with Reviewer 1 that it would be more appropriate for the paper to analyze more clinical samples but acquiring even the 38 samples was a huge task for us. In the State of Tabasco, all samples obtained from patients infected with pathogens responsible for possible pandemic diseases are strictly controlled by the local government. The local authorities agreed to provide us only with the 38 samples for this study, which was also separately approved by them.

As recommended, we added in the “Discussion” section some alternative methods to detect infections described in this study.

Reviewer 2 Report

Comments and Suggestions for Authors

The authors developed a fast POC assay for detecting ZIKV and CHIKV in patients’ infected sera. The assay can be used in endemic areas without special training, offering a diagnostic tool for monitoring outbreaks of these viruses. Portable diagnostic tools for these agents are not currently available for use in the POC setting. This system differs from conventional POC molecular tests by incorporating a nucleic acid capture membrane at the inlet of the amplification chambers, which immobilizes nucleic acid.

The authors should compare the sensitivity and specificity of the developed RT-LAMP with that of quant RT-PCR, the gold standard for such diagnosis.

Please substantiate claiming that the assay is cost-effective.

line 177: “0 mM Tris,”

Author Response

We really appreciate your time to review our manuscript. In previous data at the University of Pennsylvania, we were able to detect as few as 1 PFU of the virus with two-stage amplification, and symptomatic ZIKA patients (infected sera) present 10^3-10^6 PFU/ml when viremic, which proves that these results are equal to or even better than those of RT-PCR. This means that our two-stage amplification system can co-detect ZIKV and CHIKV in blood within a very short time with a sensitivity of 1 PFU per target per sample, comparable to the gold standard PCR, without compromising specificity.

We added the cost of this assay - according to our estimation, it is less than or equal to $10.00 per sample.

We corrected line 177.

Reviewer 3 Report

Comments and Suggestions for Authors

High sensitive molecular technique for RNA virus detection

Rozmyslowicz et al. 

The authors describe a sensitive two-step RAMP protocol for detecting multiple viruses. I suggest that the authors change the title to    Highly sensitive ….detection. This sounds better. 

In the abstract, line 19 the authors state that    we recently studied this system….This does not match with the previous sentence. Do they mean the protocol described in the manuscript?

Please check line 43.

A figure describing the protocol will be helpful. Is Fig. 4 necessary? It looks redundant with Fig. 5. A conventional PCR with the primers and the cDNA template would be good to show where the monomeric band appears.   

Line 333, one word is missing in the sentence ..The simplicity ………

Comments on the Quality of English Language

Please see the attached report.

Author Response

We really appreciate your time to review our manuscript. We changed the title following the reviewer’s suggestion. Also, we corrected line 19 in the Abstract, and line 43.

We added the figure describing the protocol and removed Fig.4 as was recommended.

We corrected line 333 as well.